# Patterns of Recurrence after Neoadjuvant Therapy in Early Breast Cancer, according to the Residual Cancer Burden Index and Reductions in Neoadjuvant Treatment Intensity

**DOI:** 10.3390/cancers13102492

**Published:** 2021-05-20

**Authors:** Christoph Suppan, Florian Posch, Hannah Deborah Mueller, Nina Mischitz, Daniel Steiner, Eva Valentina Klocker, Lisa Setaffy, Ute Bargfrieder, Robert Hammer, Hubert Hauser, Philipp J. Jost, Nadia Dandachi, Sigurd Lax, Marija Balic

**Affiliations:** 1Division of Oncology, Department of Internal Medicine, Medical University of Graz, 8036 Graz, Austria; Christoph.suppan@uniklinikum.kages.at (C.S.); florian.posch@medunigraz.at (F.P.); Hannah.mueller@medunigraz.at (H.D.M.); Nina.mischitz@stud.medunigraz.at (N.M.); Daniel.steiner@stud.medunigraz.at (D.S.); Eva.klocker@medunigraz.at (E.V.K.); Philipp.jost@medunigraz.at (P.J.J.); nadia.dandachi@medunigraz.at (N.D.); 2Department of Pathology, Hospital Graz South-West, 8020 Graz, Austria; Lisa.setaffy@kages.at (L.S.); ute.bargfrieder@kages.at (U.B.); 3Department of Surgery, Hospital Graz South-West, 8020 Graz, Austria; Robert.hammer@kages.at (R.H.); Hubert.hauser@kages.at (H.H.); 4Department of Medicine III, Klinikum rechts der Isar, TUM School of Medicine, Technical University of Munich, 81675 Munich, Germany; 5School of Medicine, Johannes Kepler University Linz, 4020 Linz, Austria

**Keywords:** early breast cancer, neoadjuvant systemic therapy, RCB

## Abstract

**Simple Summary:**

The residual cancer burden (RCB) score provides prognostic information on the survival of breast cancer patients who underwent neoadjuvant systemic therapy, with the greatest impact of higher scores on worse recurrence-free survival (RFS) and overall survival (OS) in triple-negative and HER2 positive patients. The impact of chemotherapy dose reduction on RCB is currently unknown, and should therefore be critically evaluated by clinicians. Our study confirms the prognostic relevance of the RCB score and suggests a potential association of the RCB with dose reduction having adverse impact on the RCB, thereby potentially impacting the prognosis of patients, as shown here in a large breast cancer cohort at the Medical University of Graz.

**Abstract:**

Background: The prognostic performance of the residual cancer burden (RCB) score is a promising tool for breast cancer patients undergoing neoadjuvant therapy. We independently evaluated the prognostic value of RCB scores in an extended validation cohort. Additionally, we analyzed the association between chemotherapy dose reduction and RCB scores. Methods: In this extended validation study, 367 breast cancer patients with available RCB scores were followed up for recurrence-free survival (RFS), distant disease-free survival (DDFS), and overall survival (OS). We also computed standardized cumulative doses of anthracyclines and taxanes (A/Ts) to investigate a potential interaction between neoadjuvant chemotherapy dose reduction and RCB scores. Results: Higher RCB scores were consistently associated with adverse clinical outcomes across different molecular subtypes (HR for RFS = 1.60, 95% CI 1.33–1.93, *p* < 0.0001; HR for DDFS = 1.70, 95% CI 1.39–2.05, *p* < 0.0001; HR for OS = 1.67, 95% CI 1.34–2.08, *p* < 0.0001). The adverse impact prevailed throughout 5 years of follow-up, with a peak for relapse risk between 1–2 years after surgery. Clinical outcomes of patients with RCB class 1 did not differ substantially at 5 years compared to RCB class 0. A total of 180 patients (49.1%) underwent dose reduction of neoadjuvant A/T chemotherapy. We observed a statistically significant interaction between dose reduction and higher RCB scores (interaction *p*-value = 0.042). Conclusion: Our results confirm RCB score as a prognostic marker for RFS, DDFS, and OS independent of the molecular subtype. Importantly, we show that lower doses of cumulative neoadjuvant A/T were associated with higher RCB scores in patients who required a dose reduction.

## 1. Introduction

Neoadjuvant systemic therapy (NST) is a common approach for treating early breast cancer, specifically high-risk patients. Not only does downstaging of the tumor enhance the rate of breast-conserving surgery, but due to its prognostic value, post-operative management benefits from the assessment of treatment response after NST. The pathologic complete response (pCR), generally defined as no invasive cancer in the breast or axillary lymph nodes, is a common study endpoint of neoadjuvant clinical trials associated with improved long-term outcomes and better prognosis [1,2,3].

However, the prognostic value depends on the method used for pathological assessment. Based on international recommendations, the pathological response and quantification of the extent of residual disease should be evaluated by a histological classification without showing preference for a particular one [4].

The residual cancer burden (RCB) score uses the diameter of residual disease, percentage of vital tumor cells, and diameter of the largest involved lymph node to calculate the amount of residual disease. This score has been validated with three distinct prognostic RCB classes in all breast cancer subtypes, with the most significant discriminatory power in triple-negative and Her-2 positive breast cancer. Identifying a patient subgroup with minimal residual disease (RCB I) is only one benefit of this time-saving standardized procedure. It allows estimating an excellent prognosis that is almost comparable with patients achieving a pCR or RCB 0 [5,6].

In early breast cancer, standard combination chemotherapy includes an anthracycline and a taxane, preferably in sequential treatment schedules. It has been shown that their administration leads to a one-third reduction of breast cancer mortality [7]. Several adjuvant trials have shown that increased dose intensity could improve the effectiveness of chemotherapy [8,9,10]. At the same time, dose reduction and delay of chemotherapy are associated with lower survival rates [8,11,12].

In a previous study, we validated RCB as an independent prognostic factor for breast cancer patients’ survival after neoadjuvant therapy in a single institutional study [13]. The primary objective of the present analyses was to validate these findings in a larger cohort of breast cancer patients, as measured by five-year recurrence-free survival. Furthermore, as a secondary endpoint, we evaluated the impact of chemotherapy dose reduction on the RCB.

## 2. Materials and Methods

### 2.1. Study Design

We extended our previously published validation study on the prognostic value of RCB score [13] by analyzing 367 breast cancer patients who underwent neoadjuvant treatment at the Division of Oncology, Department of Internal Medicine, Medical University of Graz between 2011 and 2020. The ethics committee of the Medical University of Graz (ethical approval number 31–212 ex 18/19) approved the study. After 43 patients (10.4%) were excluded, the final analysis cohort consisted of 367 patients in total.

The in-house electronic healthcare database and paper charts were consulted for retrieving baseline data on patient demographics, tumor characteristics, data on neoadjuvant treatment, and clinical outcome. The central registry of the Austrian Social Security Providers Association provided information on survival status. The pathologic procedure and diagnosis, including assessment of RCB score, were performed at the Department of Pathology, Hospital Graz II. Postsurgical follow-up was conducted for all patients, with regular clinical visits as well as selected imaging studies occurring quarterly during the first three years, every six months for the following two years, and yearly after that for up to a maximum of 10 years.

The standardized definitions for efficacy endpoints (STEEP) criteria were used for endpoint definition [14]. The primary endpoint of our study was five-year recurrence-free survival (RFS), defined as the date of definitive surgery until local recurrence, distant recurrence, or death from any cause, whichever occurred first during the first five years of follow-up. The secondary endpoints included five-year distant disease-free survival (DDFS) and five-year overall survival (OS). DDFS was defined as the time from definitive surgery until the date of first distant recurrence or death from any cause. OS was defined as the time from definitive surgery until death from any cause or censoring alive. To enable an analysis of neoadjuvant dose density, we computed standardized cumulative doses of the three anthracyclines used (doxorubicin (anthracycline multiplicative factor (aMF) = 1), epirubicin (aMF = 0.67), and Myocet (aMF = 1)) and the three taxanes used (paclitaxel (taxane multiplicative factor (tMF) = 1), nab-paclitaxel (tMF = 0.64), and docetaxel (tMF = 3.2)), as follows: anthracycline and taxane doses were multiplied by their respective aMFs and tMFs. The within-patient sum of these multiplied doses was then divided by the body surface area (BSA, according to the Dubois formula) of the respective patient to obtain the standardized cumulative anthracycline and taxane dose. For example, a patient with a BSA of 1.88 m^2^ who received 567 mg of epirubicin during neoadjuvant therapy will have a standardized cumulative anthracycline dose of 203 mg.

The majority of HER2+ patients (79/124, 63.7%) received dual neoadjuvant inhibition with pertuzumab and trastuzumab, and remaining patients were treated with trastuzumab only, as was the standard treatment at the time of administration.

### 2.2. Pathology and RCB Evaluation

Histological diagnosis was performed on pre-therapeutic core needle biopsies and included assessment of ER, PR, HER2, and Ki67 labeling index according to the WHO guidelines, as previously described [13]. Post-therapeutic ypTNM classification and RCB were assessed on the surgical specimens. The RCB score was assessed using the RCB calculator on the MD Anderson website, and the RCB score and RCB class were reported accordingly [5,6,15].

### 2.3. Statistical Methods

Stata 16.1 by ND and FP (Stata Corp., Houston, TX, United States) was used in all statistical analyses. Continuous variables were reported as medians (25th–75th percentile) and count data as absolute numbers (%). Rank-sum tests were used to compare how continuous variables were distributed between two groups, whereas χ^2^-tests and Fisher’s exact tests were used for investigating the association between two categorical variables. The association between covariates and the continuous RCB score was examined using fitted simple and multiple linear regression models, including models with dose densities of taxanes and anthracyclines, as well as the interaction of the two. A multivariable linear regression model of the RCB score was obtained by a backward elimination algorithm, with a *p* for exclusion of 0.05. The reverse Kaplan–Meier estimator [16] was used in estimating median follow-up time. Kaplan–Meier estimators were used in comparing RFS and OS, and their functions were compared with log-rank tests between two or more groups. Uni- and multivariable Cox proportional hazards models were used to calculate the association between prognostic variables, such as the RCB scores and classes, respectively, and the risk of recurrence or death. Time-dependent recurrence rate curves, according to RCB class, were generated with Royston–Parmar models under proportional hazards (Schonfeld test for RCB-class *p* = 0.315; Stata routine stpm2) [17].

## 3. Results

### 3.1. Patient Characteristics

This expanded validation study analyzed 367 breast cancer patients with available RCB scores treated at the Division of Oncology between 2011 and 2020 (Figure 1). Patient characteristics of this study cohort are summarized in Table 1. At the start of neoadjuvant treatment, the median patient age was 54 years (5th–75th percentile: 47–63). All but two patients were female (*n* = 365, 99.5%), and the majority of the patients had grade 3 tumors (*n* = 234, 65.4%) with a median Ki67 of 40% (25th–75th percentile: 28–70; range 5–95). One-third (*n* = 127 (34.6%)) of the carcinomas were HER2-amplified, and 108 (29.4%) were triple-negative. All patients received a sequential anthracycline–taxane-based neoadjuvant treatment regimen, and 124 patients (33.8%) additionally received HER2-directed treatment in the neoadjuvant setting. Breast-conserving surgery was performed in 255 patients (69.5%) and mastectomies in 112 patients (30.5%). After neoadjuvant treatment, axillary lymph node dissection was performed in 241 patients (65.7%) and sentinel node biopsy in 126 patients (34.3%).

### 3.2. Association of RCB Score with Clinical Outcome

During a median follow-up time (truncated at five years) of 4.1 years (95% CI: 3.8–4.5), we observed 60 RFS events, 56 DDFS events, and 43 OS events. Individually, we observed 11 local recurrences and 48 distant recurrences, and 48 patients died. The estimated five-year RFS, DDFS, and OS rates for the entire study population were 73% (95% CI: 66–79), 76% (95% CI: 68–81), and 83% (95% CI: 77–87), respectively.

Compared to patients who remained free from an RFS event, patients with an RFS event during follow-up had a significantly higher prevalence of mastectomy and ALND, as well as a significantly larger post-neoadjuvant tumor size and higher post-neoadjuvant nodal stage. The median RCB score was considerably higher in patients with an RFS event during follow-up than patients who remained event-free (median RCB score: 2.22 versus 1.33, rank-sum *p* < 0.0001).

In univariable COX regression, higher RCB scores were associated with worse RFS (HR for recurrence or death per one-point increase = 1.60, 95% CI 1.33–1.93, *p* < 0.0001), worse DDFS (HR per one-point increase = 1.70, 95% CI 1.39–2.05, *p* < 0.0001), and worse OS (HR per one-point increase = 1.67, 95% CI 1.34–2.08, *p* < 0.0001). Similarly, a higher RCB class was also significantly associated with worse RFS, DDFS, and OS (Table 2 and Figure 2). The clinical outcome of patients with RCB class 1 did not differ substantially at 5 years compared to patients with RCB class 0 (Figure 2). The strongest other univariable predictors of worse clinical outcomes were residual tumor and residual axillary lymph node metastases (yTNM categories), both components of the RCB score (Table 2).

Evidence for non-proportionality of hazards, according to the RCB score (interaction HR for RFS with linear follow-up time = 1.063, *p* = 0.406) was not observed, which suggests that the adverse prognostic impact of RCB score on RFS prevails throughout the follow-up period. Using flexible parametric survival modeling under the PH assumption, the estimated recurrence rate for patients with RCB class 3 tumors was significantly increased throughout follow-up time, and reached a peak between 1 and 2 years after definitive surgery. This peak was less prominent in patients with RCB class 1 and 2 tumors. Consistent with the Kaplan–Meier analysis, the recurrence risk was constantly low throughout five years of follow-up in patients with RCB class 0 tumors (Figure 3A).

In contrast, the proportional hazard assumptions were not met for tumor subtype (Schoenfeld test *p*-value = 0.009). In flexible parametric modeling, patients with triple-negative tumors had the highest recurrence peak within one year, followed by a decline over the five-year follow-up time. Patients with HER2+ tumors revealed a lower but constantly elevated recurrence rate, while the recurrence rate for HR+/HER2- tumors showed a constant increase over the entire follow-up period (Figure 3B).

Finally, the adverse impact of RCB score on RFS was investigated, with specific attention to whether the values differ across selected clinically important subgroups. An association found between higher RCB scores and worse RFS was consistent across different molecular subtypes (interaction *p*-value = 0.155), for tumor grade (interaction *p*-value = 0.673), for patient age (interaction *p*-value = 0.537), and type of definitive surgery (interaction *p*-value = 0.553). Although the interaction test between RCB score and molecular subtype was not significant (*p* = 0.155), we found that the magnitude of the adverse prognostic impact of increasing RCB score on RFS was weaker, and did not reach significance in the HR+/HER2- subtype (HR = 1.44, 95% CI: 0.99–2.08, *p* = 0.056) compared to the HER2+ subtype (HR = 1.71, 95% CI: 1.20–2.44, *p* = 0.003) and the triple-negative subtype (HR = 2.14, 95% CI: 1.55–2.95, *p* < 0.001). This is in line with the distribution of RCB score varying between the subtypes, as shown in Appendix A.

However, we found that the magnitude of the adverse prognostic impact of RCB scores on RFS significantly differed by Ki67 labelling index, with a significantly greater association observed in patients with high Ki67 labelling index (interaction *p*-value = 0.010). This implies that residual tumors with high proliferation indices represent a particularly poor prognostic subtype.

### 3.3. Analysis of RCB Score and Association of A/T Dose Reduction with RCB Score and Clinical Outcome

In linear regression analysis, the strongest univariable predictors of low RCB score were triple-negative and HER2-positive subtypes, a tumor grade of G3, and a higher Ki67 labelling index (Table 3). In a final multivariable regression model, only the breast cancer subtype and Ki67 labelling index remained independent predictors of RCB (multivariable model #1 in Table 3).

In an exploratory analysis, we investigated whether dose reduction of neoadjuvant A/T treatment was associated with higher RCB scores. Overall, 180 patients (49.1%) underwent dose reduction of neoadjuvant A/T-based chemotherapy. Consistent with our hypothesis, an interaction term of A/T doses by dose reduction was significant in multiple linear regression (interaction *p*-value = 0.047, multivariable model #2 in Table 3), indicating that potentially adverse effects of dose reductions on neoadjuvant treatment response are less pronounced given that the magnitude of dose reduction is small, and vice versa. In multivariable linear regression that included molecular subtype and Ki67 labelling index, the cumulative A/T doses by dose reduction interaction term remained statistically significant (interaction *p*-value = 0.042, multivariable model #3 in Table 3 and Figure 4).

## 4. Discussion

In the present study, we confirm the prognostic value of the RCB in a larger cohort of our patients, as well as an extent observation period. In addition to this internal confirmation, our data support the approach of adherence to dose and schedule of neoadjuvant treatment by implicating that dose reduction for any reason may impact the RCB and therefore prognosis of patients. Thus, dose reduction should be considered carefully.

With additional data since our previous report [13], we evaluated the prognostic impact of RCB scoring and classification in an extended population cohort of breast cancer patients undergoing NST at our department. Our cohort represents a typical neoadjuvant population, consisting of high-risk patients with either triple-negative, HER-2-positive, or HR-positive breast cancer with unfavorable features. In the overall group, the median Ki67 was 40%, and 60% had G3 differentiated tumors. These features are comparable with the RCB validation cohort published by Symmans et al. [6]. Breast cancer patients with these features also tend to gain higher pCR rates by neoadjuvant chemotherapy, while being at greater risk of recurrence in general [18]. The better responsiveness of these patients to NST not only correlates with the greater vulnerability of more aggressive tumors to chemotherapy, but also with the addition of specific targeted therapies or immunotherapies when indicated. Defined by more aggressive biology, these subtypes have been repeatedly associated with a more prolonged RFS and OS when pCR is diagnosed after surgery [19]. Additional and more differentiated information, in particular in patients without pCR but a similar prognostic impact as pCR, has been improved by the inclusion of RCB scoring in response evaluation [19,20]. We show that known components of RCB score, such as post-neoadjuvant tumor size, nodal status, and the number of positive lymph nodes, were significantly associated with worse RFS, comparable to published data [20,21,22].

Our updated results reveal that the prognostic effect of RCB on RFS and OS is mainly driven by higher RCB classes II and III, which might also be influenced by the relatively small number of patients that we were able to analyze. The fact that the differentiation is less likely in classes 0 and I may also be correlated with the size of the population, as shown in the last presentation of the largest population analyzed by the RCB score so far. Specifically, Yau et al. demonstrated in a large population of 5100 patients that even a separation of RCB 0 (69%) and I (11%) was associated with adverse impact on the prognosis of the latter [23]. One limitation of our study is the short follow-up of five years. This limitation is especially relevant because HR-positive breast cancer patients tend to recur later. In contrast, patients with triple-negative tumors have the highest risk of recurrence in the first three to five years after surgery [24]. On the other hand, particularly in the groups of patients with adverse biology like being triple-negative, as we were able to confirm in the present study, or having HER-2-positive tumors, the major proportion of recurrences occur in this time period. The benefit of chemotherapy is mostly seen in the first three years after initial treatment, considering that HR-positive patients often benefit from prolonged administration of endocrine therapy [25]. For HR-positive patients there is no definitive conclusion. Whereas Symmans et al. demonstrated prognostic significance in HR-positive patients [6], Hamy et al. were not able to demonstrate this prognostic effect [26], and in our own analyses, the significance was borderline. This data confirms the need for a larger number of patients and longer follow-up in the HR-positive group, beyond the follow up time provided here. About 45% of the patients in our cohort were diagnosed with an RCB score of 0 or 1, whereas 39% had an RCB score of 2. Notably, this information about residual disease further impacts the post-neoadjuvant treatment of patients, particularly in triple-negative patients [27,28] and HER2-positive patients. The phase 3 KATHERINE trial tested 14 cycles of trastuzumab versus trastuzumab emtansine (T-DM1), an antibody–drug conjugate adjuvant therapy in HER2-positive breast cancer patients with residual disease after neoadjuvant treatment. Invasive disease-free survival was significantly higher in the T-DM1 group (HR = 0.5). This led to implementing a new standard of care treatment for patients with residual disease [29]. In triple-negative breast cancer, residual disease after neoadjuvant chemotherapy can be treated with capecitabine. The overall survival rate could be increased from 70.3% to 78.8% (HR = 0.52) by additional chemotherapy [27]. Since this data was published, clinicians tend to discuss post-neoadjuvant capecitabine with their patients, especially in RCB II or III cases.

As an exploratory endpoint, we evaluated the impact of chemotherapy dose reduction on the RCB score. We found an association of dose reduction of anthracycline and taxane-based chemotherapy with higher RCB scores. Not only the optimal dosage, but also the timing of these drugs in breast cancer treatment are unclear. More than 30 years ago, the dose intensity of adjuvant chemotherapy in breast cancer was described as an independently significant correlate of relapse-free survival [10]. A meta-analysis of over 37,000 patients showed a reduction of recurrence risk and death from breast cancer by increasing the dose intensity [9]. This can either be achieved by administering a higher single dose or shortening the intervals between the treatment cycles to gain more dose-density [30]. The log-cell kill hypothesis states that a given dose of chemotherapy will always kill a constant fraction of the tumor, regardless of the tumor size or the number of cells. Therefore, a higher dose of chemotherapy is supposed to eliminate a larger fraction of tumor cells. However, the effectiveness of tumor-specific therapy is also associated with the rate of tumor regrowth between the treatment cycles, suggesting that shorter treatment intervals also play an essential role [30,31]. This hypothesis is supported by Henderson et al., who described no benefit to solely escalating the anthracycline dose, while others found a lesser benefit with doses below the threshold of the current standard [11,32]. Dose reduction in chemotherapy patients is common, and is performed in about 50% of all patients in daily practice, mostly because of hematologic toxicities or for no objective medical reason [33,34]. However, reducing doses below the accepted conventional threshold should be avoided, and may cause an inferior outcome [11,12,33]. In our exploratory analysis, we showed that 49.1% of patients received a dose reduction. A significant interaction indicated that the potential adverse effects of dose reductions on neoadjuvant treatment response might depend on the magnitude of dose reduction. This observation is preliminary and based on an analysis of heterogeneously administered neoadjuvant regimes, age of the patient, dose intensity, and density, but provides potentially important insights. Such investigations should be performed in a clinical trial with homogeneously treated patients, as well as carefully observed and documented dose reduction and its reasons, in order to address the clinical relevance from both aspects.

## 5. Conclusions

In summary, with this study, we confirmed the prognostic value of RCB scoring in our institutions as a feasible approach to identify patients at higher risk for relapse. We further provide preliminary data on the importance of careful consideration of dose reduction and interruption of NST. We support the clinical implementation of RCB scores for selecting patients for intensified treatment and the generation of risk-adapted follow-up of higher classes. Finally, we emphasize the importance of administrating scheduled therapies to improve the prognosis of NST-treated patients.

## Figures and Tables

**Figure 1 cancers-13-02492-f001:**
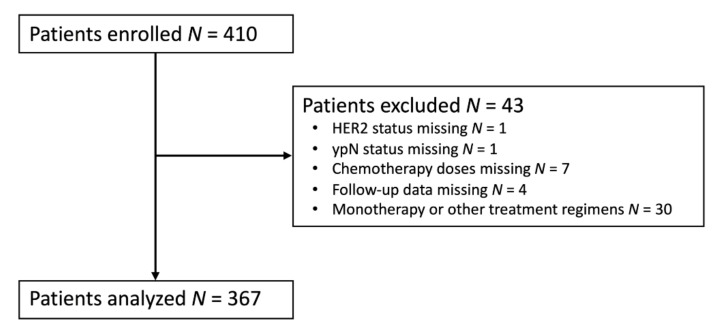
Consort diagram for the study showing number of patients included and reasons for exclusion.

**Figure 2 cancers-13-02492-f002:**
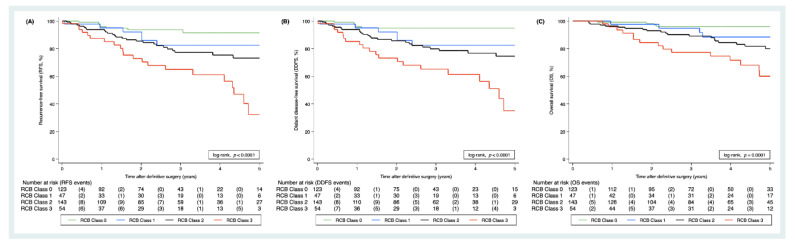
Kaplan–Meier RFS (**A**), DDFS (**B**), and OS (**C**) functions by RCB class. The numbers below the Kaplan–Meier plot form a risk table, whereas the round brackets contain the number of events occurring within the respective report.

**Figure 3 cancers-13-02492-f003:**
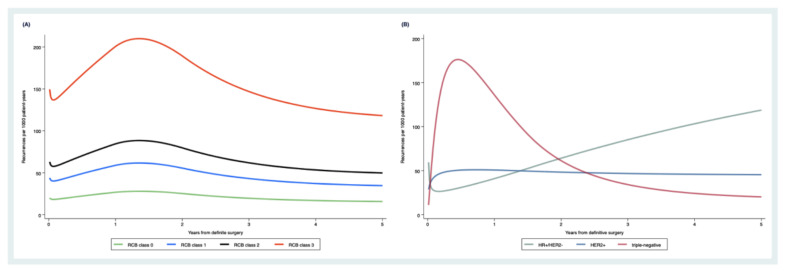
Estimated recurrence rates within 5 years of definitive surgery, according to RCB class (**A**) and subtype (**B**). Rate curves were predicted with a flexible parametric survival model on the log(cumulative hazard) scale, allowing subtype to vary by time since definitive surgery.

**Figure 4 cancers-13-02492-f004:**
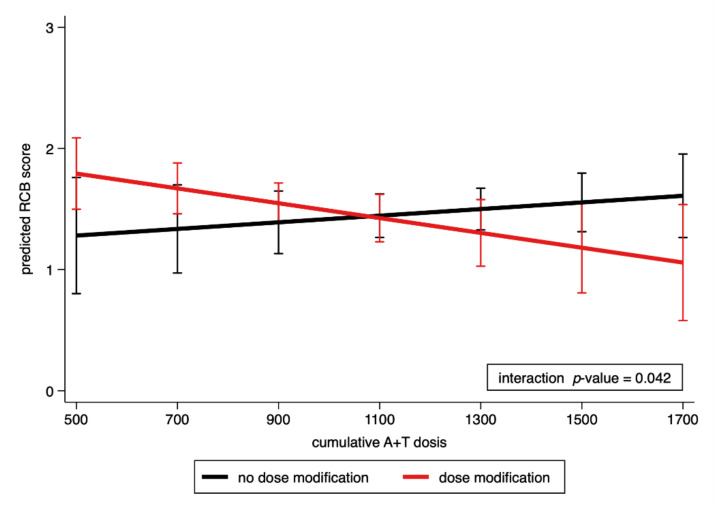
Interaction of cumulative A + T doses by dose reduction. Lower doses of cumulative neoadjuvant A/T are associated with higher RCB scores in patients who required a dose reduction.

**Table 1 cancers-13-02492-t001:** Baseline characteristics overall and by RFS event status.

	*n*	Total	No RFS Event during Follow-Up	RFS Event during Follow-Up	*p*-Value
	(% miss.)	*n* = 367	*n* = 307	*n* = 60	
Age at neoadjuvant treatment start (years)	367 (0)	54.6 (47.0–63.3)	54.7 (47.5–63.4)	53.4 (45.0–59.5)	0.149
Female gender	367 (0)	365 (99.5%)	306 (99.7%)	59 (98.3%)	0.301
Molecular breast cancer subtype	367 (0)				0.146
HR-positive/HER2-		132 (36.0%)	110 (35.8%)	22 (36.7%)	
HER2+		127 (34.6%)	112 (36.5%)	15 (25.0%)	
Triple-negative		108 (29.4%)	85 (27.7%)	23 (38.3%)	
Histological grade	358 (2.5)				0.349
G1		3 (0.8%)	3 (1.0%)	0 (0.0%)	
G2		121 (33.8%)	106 (35.2%)	15 (26.3%)	
G3		234 (65.4%)	192 (63.8%)	42 (73.7%)	
Ki67 labeling index (%)	366 (0.3)	40.0 (27.5–70.0)	40.0 (25.0–70.0)	40.0 (30.0–70.0)	0.357
Surgical outcome	367 (0)				0.040
Mastectomy		112 (30.5%)	87 (28.3%)	25 (41.7%)	
Breast conservation		255 (69.5%)	220 (71.7%)	35 (58.3%)	
Definitive axillary procedure	367 (0)				<0.001
Sentinel node biopsy (SNB)		126 (34.3%)	118 (38.4%)	8 (13.3%)	
Axillary lymph node dissection (ALND)		241 (65.7%)	189 (61.6%)	52 (86.7%)	
Post-neoadjuvant tumor category (ypT)	367 (0)				<0.001
ypTis-ypT0		140 (38.1%)	127 (41.4%)	13 (21.7%)	
ypT1		157 (42.8%)	132 (43.0%)	25 (41.7%)	
ypT2		51 (13.9%)	37 (12.0%)	14 (23.3%)	
ypT3-ypT4		19 (5.2%)	11 (3.6%)	8 (13.3%)	
Post-neoadjuvant nodal status (ypN)	367 (0)				<0.001
ypN0		265 (72.2%)	235 (76.6%)	30 (50.0%)	
ypN1		64 (17.4%)	51 (16.6%)	13 (21.7%)	
ypN2		32 (8.7%)	19 (6.2%)	13 (21.7%)	
ypN3		6 (1.6%)	2 (0.7%)	4 (6.6%)	
Number of positive nodes	367 (0)	0.0 (0.0–1.0)	0.0 (0.0–0.0)	1.0 (0.0–4.0)	<0.001
Adjuvant endocrine therapy	367 (0)	191 (52.0%)	162 (52.8%)	29 (48.3%)	0.529
Adjuvant chemotherapy ± anti-HER2	367 (0)	145 (39.5%)	126 (41.0%)	19 (31.7%)	0.174
RCB score	367 (0)	1.52 (0.00–2.34)	1.33 (0.00–2.10)	2.21 (1.54–3.60)	<0.001
RCB class	367 (0)				<0.001
RCB Class 0		123 (33.5%)	116 (37.8%)	7 (11.7%)	
RCB Class 1		47 (12.8%)	41 (13.4%)	6 (10.0%)	
RCB Class 2		143 (39.0%)	117 (38.1%)	26 (43.3%)	
RCB Class 3		54 (14.7%)	33 (10.7%)	21 (35.0%)	

ER: estrogen receptor, HER2: human epidermal growth factor receptor 2 (erb-B2), RFS: recurrence-free survival. Data are reported as medians (25th–75th percentile) for continuous variables and absolute frequencies (%) for count data. *p*-values are from Wilcoxon’s rank-sum tests χ^2^ tests or Fisher’s exact tests) Adjuvant anti-HER2 therapy included seven patients with T-DM1, five patients with trastuzumab and pertuzumab, and all other patients had trastuzumab only. Total duration was one year, including neoadjuvant cycles.

**Table 2 cancers-13-02492-t002:** Univariable predictors of five-year RFS, DDFS, and OS in the study cohort (*n* = 367).

Variable	RFS (Events = 60)			DDFS (Events = 56)			OS (Events = 43)		
	HR	95% CI	*p*-Value	HR	95% CI	*p*-Value	HR	95% CI	*p*-Value
RCB score (by one-point increase)	1.60	1.33–1.93	<0.0001	1.70	1.39–2.05	<0.0001	1.67	1.34–2.08	<0.0001
RCB class									
RCB Class 0	Ref.	Ref.	Ref.	Ref.	Ref.	Ref.	Ref.	Ref.	Ref.
RCB Class 1	2.18	0.73–6.50	0.161	3.11	0.95–10.19	0.061	2.53	0.63–10.12	0.189
RCB Class 2	3.15	1.37–7.27	0.007	4.23	1.62–11.07	0.003	4.23	1.47–12.55	0.008
RCB Class 3	7.44	3.16–17.50	<0.0001	10.23	3.84–27.25	<0.0001	9.13	3.03–27.51	<0.0001
Age at treatment start (per five-year increase)	0.92	0.82–1.03	0.134	0.92	0.82–1.04	0.166	0.88	0.77–1.01	0.065
Molecular breast cancer subtype									
HR-positive/HER2-	Ref.	Ref.	Ref.	Ref.	Ref.	Ref.	Ref.	Ref.	Ref.
HER2+	0.78	0.40–1.50	0.454	0.74	0.36–1.48	0.385	0.86	0.38–1.97	0.727
Triple-negative	1.46	0.82–2.63	0.202	1.58	0.87–2.88	0.135	2.08	1.04–4.19	0.040
Tumor grade G3	1.35	0.75–2.43	0.321	1.69	0.89–3.21	0.108	1.38	0.70–2.75	0.353
Ki67 (per 10% increase)	1.05	0.94–1.17	0.392	1.07	0.96–1.20	0.215	1.10	0.97–1.25	0.137
Breast conservation	0.59	0.36–0.99	0.046	0.61	0.36–1.03	0.065	0.67	0.36–1.23	0.199
Axillary lymph node dissection	2.38	1.13–5.03	0.023	2.66	1.20–5.89	0.016	3.60	1.28–10.11	0.015
Post-neoadjuvant tumor category (ypT)									
ypTis-ypT0	Ref.	Ref.	Ref.	Ref.	Ref.	Ref.	Ref.	Ref.	Ref.
ypT1	1.67	0.85–3.26	0.135	1.91	0.93–3.90	0.076	2.12	0.93–4.84	0.075
ypT2	3.10	1.46–6.60	0.003	3.40	1.52–7.59	0.003	3.72	1.47–9.43	0.006
ypT3-ypT4	4.95	2.05–11.95	<0.0001	6.35	2.55–15.80	<0.0001	5.63	1.95–16.25	0.001
Number of positive nodes (per 1 increase)	1.15	1.09–1.20	<0.0001	1.16	1.10–1.21	<0.0001	1.15	1.08–1.22	<0.0001
Post-neoadjuvant nodal status (ypN)									
ypN0	Ref.	Ref.	Ref.	Ref.	Ref.	Ref.	Ref.	Ref.	Ref.
ypN1	1.86	0.97–3.56	0.062	1.91	0.97–3.77	0.062	1.77	0.81–3.84	0.150
ypN2	4.20	2.19–8.06	<0.0001	4.67	2.41–9.06	<0.0001	4.40	2.13–9.08	<0.0001
ypN3	7.37	2.59–20.98	<0.0001	11.01	3.83–31.67	<0.0001	2.47	0.33–18.30	0.378
Adjuvant endocrine therapy	0.81	0.49–1.34	0.406	0.88	0.52–1.48	0.627	0.63	0.34–1.15	0.132
Adjuvant chemotherapy ± anti-HER2	0.83	0.48–1.42	0.489	0.77	0.43–1.35	0.359	0.84	0.44–1.58	0.584
NAC dose modification	1.25	0.75–2.08	0.392	1.16	0.68–1.97	0.584	1.18	0.65–2.15	0.587
Cumulative A/T doses (per 100 units increase)	0.99	0.91–1.06	0.713	1.02	0.95–1.11	0.534	1.01	0.93–1.1	0.793

ER: estrogen receptor, HER2: human epidermal growth factor receptor 2 (erb-B2), CI: confidence interval, Ref.: reference group, RFS: recurrence-free survival, DDFS: distant disease-free survival, OS: overall survival, A/T: anthracycline/taxane.

**Table 3 cancers-13-02492-t003:** Univariable and multivariable regression models investigating predictors of the RCB score.

Models	Variable	Regression Coefficient β	95% CI	*p*-Value
Univariable models	Age at treatment start (per five-year increase)	0.04	−0.02–0.10	0.179
	Molecular subtype			
	HR+	Ref.	Ref.	Ref.
	HER2+	−1.40	−1.69 to (−1.10)	<0.0001
	Triple-negative	−1.07	−1.38 to (−0.77)	<0.0001
	Tumor grade G3	−0.79	−1.07 to (−0.51)	<0.0001
	Ki67 index (per 10% increase)	−0.17	−0.22 to (−0.11)	<0.0001
	Dose modification	0.06	−0.22–0.33	0.689
	Cumulative A + T dose (per 100 units increase)	−0.01	−0.06–0.03	0.532
Multi-variable model #1	Molecular subtype			
	HR+	Ref.	Ref.	Ref.
	HER2+	−1.41	−1.68 to (−1.13)	<0.0001
	Triple-negative	−0.67	−0.99 to (−0.36)	<0.0001
	Ki67 index (per 10% increase)	−0.17	−0.23 to (−0.12)	<0.0001
Multi-variable model #2	Dose modification	1.11	−0.01–2.24	0.053
	Cumulative A + T dose (per 100 units increase)	0.04	−0.03–0.12	0.262
	Dose modification ^#^ cumulative A + T dose ^a^	−0.10	−0.20–0.00	0.047
Multi-variable model #3	Molecular subtype			
	HR+	Ref.	Ref.	Ref.
	HER2+	−1.42	−1.69 to (−1.14)	<0.0001
	Triple-negative	−0.68	−1.00 to (−0.36)	<0.0001
	Ki67 index (per 10% increase)	−0.17	−0.23 to (−0.11)	<0.0001
	Dose modification	0.95	−0.01–1.92	0.052
	Cumulative A + T dose (per 100 units increase)	0.03	−0.05–0.09	0.392
	Dose modification ^#^ cumulative A + T dose ^a^	−0.09	−0.17–0.00	0.042

HR: hormone receptor, HER2: human epidermal growth factor receptor 2 (erb-B2), CI: confidence interval, Ref.: reference group, ^a,#^ interaction term in the model.

## Data Availability

The data are not publicly available and cannot be shared under the current ethics committee approval.

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
