# Peer review of "Patterns of Recurrence after Neoadjuvant Therapy in Early Breast Cancer, according to the Residual Cancer Burden Index and Reductions in Neoadjuvant Treatment Intensity"

_cancers, 2021, doi:10.3390/cancers13102492_

Round 1
Reviewer 1 Report
The study aims to confirm the validation (on a retrospective series of 367 patients) of the prognostic performance of the Residual Cancer Burden in early breast cancer as well as evaluate the impact of chemotherapy dose reduction on the RCB.
here are several critical issues regarding:
1) external validation (?)
“We externally evaluated the prognostic value of RCB scores in an extended validation cohort “. page 1 line 23
“The pathologic procedure and diagnosis, including assessment of RCB score, were performed at the Department of Pathology, Hospital Graz II”. Page2 lines 86-87
2) the selection of the cases
“Our cohort represents a typical neoadjuvant 236 population consisting of high-risk patients with either triple-negative, HER-2-positive, or 237 HR-positive breast cancer with unfavorable features. “ page 8 line 238
although the reported Ki 67 values are never lower than 25, these are not under-stratification based on the different molecular types. This data is relevant in a retrospective series in which 132/ 367 patients are RE + (HER2-) and 127 are G1-G2. The percentage of potentially hormone-sensitive patients should be specified in a small retrospective series. In this perspective, it would be interesting to have information on the level of expression of both hormone receptors as well, in light of the rate of 70/367 patients with residual disease> T2 (ypT2 + ypT3 + ypT4), 38/367 ypN2-3.
Hormone dependence is also demonstrated by the number of patients undergoing adjuvant hormone therapy (191/367…. they were not 132?)
Two male breast cancer included?
3) the duration of the follow-up
the authors included 184 patients (30 patients / year) from 2011 to 2016 (reference 13). Therefore 183 patients were included from 2017 to 2020 (45 patients / year) suggesting that at least 90 patients can have a follow-up of less than 2 years.
“We extended our previously published validation study on the prognostic value of RCB score (13)” page 2 line 77
“between 2011 and 2020”. page 2 line 80
why (if the duration of the follow-up is 10 years) the authors have to collected the mortality data from the central registry of the Austrian Social Security Providers Association?
“Information on survival status was collected from the central registry of the Austrian Social Security Providers Association”. Page 2 line 85
“All patients underwent postsurgical follow-up, with regular clinical visits and selected imaging studies every three months during the first three years, every six months during the subsequent two years, and yearly after that for a maximum duration of 10 years”. Page 2 lines 88-91
“During a median follow-up time (truncated at five years) of 4.1 years” page 5 line 157
4) chemotherapy in adjuvant arm (145/367) (only recently become clinical practice in TNBC and HER2 + tumors that do not reach pCR)
5) the very high rate of dose reduction even in an un-selected retrospective serie (the authors themselves prove to be detrimental)
Author Response
Please find a point by point response in the attached word document.

Reviewer 2 Report
In this clinical study Suppan et al have analyzed the residual breast tumor burden after neoadjuvant therapy.
Correlative analyses clearly show that higher residual cancer burden (RCB) score is clearly associated with worse clinical outcome (RFS, DDFS and OS). Although this finding is not new, the confirmation of this association in this large study encompassing 367 breast cancer patients further emphasizes the prognostic value of RCB score after neoadjuvant chemotherapy in breast cancer.
Additionally, the authors report a significant interaction between cumulative dose redaction and RCB score. This is also concordant with previous studies showing that dose reduction is linked with inferior survival rates.
Although the novelty of the work is somewhat limited, the manuscript presents a large clinical study that is conducted rigorously and provides important and pertinent clinical information; and therefore could be considered for publication.
Author Response
We thank reveiwer #2 for this very positive comment.
Reviewer 3 Report
Thank you for asking me to review this manuscript which evaluates the prognostic performance of the RCB score following neoadjuvant chemotherapy in a cohort of patients receiving neoadjuvant chemotherapy, and furthermore evaluates the association between chemotherapy dose reduction and RCB.
This study adds to published data showing the prognostic value of RCB - in particular, the Symmans paper of 2017 which demonstrates RCB to be prognostic for long-term survival after neoadjuvant therapy in all subtypes (HR+ HER2-, HER2+ and TNBC). In contrast, a more recent French study (Hamy et al, PLOS ONE 2020) found RCB to be prognostic in only HER2+ and TNBC, but not HR+/HER2-ve disease. Both the Symmans series and the Curie paper have considerably longer follow up than the cohort presented here, and both analyse the performance of RCB by subtype, as does the pooled meta-analysis (Yau et al) presented at SABCS and cited in the discussion here.
The authors should consider presenting their data on the prognostic performance of the RCB by subtype in this series. At the very least, the distribution of RCB scores should be shown by subtype. It would be interesting to see whether there were higher RCB scores in HR+/HER2- disease as this subtypes generally appears to have lower pCR rates following neoadjuvant chemotherapy. If so, the relatively short, 5 year follow up in this series (correctly acknowledged by the authors as a limitation) may not be long enough to identify late recurrence in the HR+ population. Presumably HR+ patients received adjuvant endocrine therapy although this is not mentioned - can the authors comment?
There is a peak of recurrence seen in the RCB class 3 patients between 1-2 years after surgery. Was this driven by any particular subtype of tumours as it appears to correspond to the early peak in recurrence seen in TNBC?
There are a few additional, relatively minor points that should be considered. Did the HER2+ve patients receive single or dual agent neoadjuvant anti-HER2 therapy? Clearly this would be expected to influence pCR rates. What adjuvant anti-HER2 targeted therapy was given to these patients? Did TNBC patients with RCB >0 receive adjuvant capecitabine? The evidence for adjuvant capecitabine in this context is equivocal (see Lluch et al, JCO 2020) and this GEICAM study should be cited in addition to the CREATE-X study (Masuda et al, reference 6) in the discussion.
In terms of the dose reduction data - can the authors clarify whether dose reductions related to a reduction in the dose given at each cycle, or whether some patients had fewer cycles of treatment than planned prior to surgery? If so, did any of the patients with either a dose reductino or a reduced number of cycles receive adjuvant systemic therapy?
Author Response
Please find a point by point review in the attached word document.

Round 2
Reviewer 1 Report
accept in present form
Author Response
Thank you for your positive answer!
Reviewer 3 Report
Thank you for considering my previous comments, which have largely been addressed by this revision. I have a couple of minor remaining queries only:
Re the HER2+ve patient cohort - thank you for providing information on their neoadjuvant treatment in respect of single or dual anti-HER2 therapies. As per my previous comment could you also please clarify what adjuvant anti-HER2 therapy was given (did all patients get 18 cycles trastuzumab total; was any adjuvant pertuzumab used in the latter part of the study)?
An additional row has been added to table 1 - adjuvant chemotherapy and/or immunotherapy (and there is a typographical error in immunotherapy) - this is the only mention of immunotherapy that I can see in the manuscript. What immunotherapy was given and to which patients?
With regard to the sentence:
"...the magnitude of the adverse prognostic impact of increasing RCB score on RFS was weaker and only borderline significant in the HR+/HER2- subtype (HR 1.44, 95%CI 0.99-2.08, p=0.056"
Was a p value of <0.05 considered significant? In which case this should say "did not reach significance" rather than "borderline significant"
Author Response
Dear reviewer,
Thank you for your confirmative answer. Please find remaining answers in the attached file.
Best regards,
Marija Balic on behalf of all coauthors
